# Incidence of long-term conditions in the Latin American community of London: A validation and retrospective cohort study of 890,922 primary care records, 2005–2022

**James Scuffell** [1]*, **James Bailey**[2], **Hiten Dodhia** [3], **Stevo Durbaba** [1], **Mark Ashworth**[1]

1 Department of Population Health Sciences, School of Life Course and Population Sciences, King's College London, London, United Kingdom, 2 Research Department of Primary Care and Population Health, University College London, London, United Kingdom, 3 Public Health Department, London Borough of Lambeth, Brixton, London, United Kingdom

* James.scuffell@kcl.ac.uk

## Abstract

### Background

Minoritised populations in the United Kingdom frequently identify in multiple ethnic groupings and therefore little is known of their health needs. There were 136,062 Latin American people recorded in the 2021 UK Census across six different ethnic groups.

### Aim

Characterise the incidence of long-term conditions (LTCs) and multiple LTCs (mLTCs) amongst the Latin American community of London. Compare the incidence of LTCs in the Latin American population to other ethnic groups.

### Design and setting

Retrospective cohort study using pseudonymised primary care data from 890,922 individuals in an urban, superdiverse area of London from 2005–2022.

### Method

Latin American individuals were identified using country of birth, language and ethnicity codes, and validated against Census findings. Multivariable competing risks regression models estimated the effect of being Latin American, compared to the White British ethnic group, on incidence of 32 LTCs and risk factors relevant to urban populations.

### Results

28,617 Latin American people were identified in this cohort, 3.2% of total. In multivariable analysis, compared to the White British ethnic group, being Latin American was associated with twice the rate of HIV/AIDS (hazard ratio (HR) 2.00; 95% confidence interval (CI) 1.65–2.43), 60% increased rate of diabetes (HR 1.61; 95%CI 1.47–1.77) and almost twice the

**Data Availability Statement:** The data are not publicly available to share because they contain

pseudonymised patient identifiable information. The research group can provide descriptive aggregate data. Requests should be made to Mark Ashworth, the Lambeth DataNet lead for King's College London, responsible for approving data access (mark.ashworth@kcl.ac.uk).

**Funding:** JS is funded by a National Institute for Health Research In-Practice Fellowship (NIHR303520). JB is funded by a National Institute for Health Research Doctoral Research Fellowship (NIHR302551). The funders had no role in study design, data collection and analysis, decision to publish, or preparation of the manuscript. The views expressed are those of the author(s) and not necessarily those of the NIHR or the Department of Health and Social Care.

**Competing interests:** The authors have declared that no competing interests exist.

rate of systemic lupus erythematosus and rheumatoid arthritis (HRs 2.28; 95% CI 1.18–4.38 and 1.69; 95% CI 1.32–2.17 respectively).

## Conclusion

Using commonly-recorded primary care codes accurately and reliably identifies markedly higher risks of HIV/AIDS, diabetes and joint disease among London's Latin American population. These data can be used to target inclusive and equitable health interventions.

## Introduction

The United Kingdom (UK) Census names eighteen ethnic groups in five higher levels: Asian, Black, Mixed, Other and White [1]. However in 2021 7.5% of those of White ethnicity, and over a quarter of Mixed and Multiple ethnicity respondents, identified in 'other' ethnic groups that did not resolve to these eighteen categories [2].

Minoritised populations in the UK are less likely to access preventative care and have lower confidence in managing health conditions [3]. Compared to the White British group, Other ethnic groups are less likely to have a recent recorded blood pressure [4] or HbA1c test [5]. They are also less likely to be tested for COVID-19 [5] or take up NHS Health Checks [6]. National data also demonstrate people of Other ethnicities report much lower confidence with self-care and feel less supported in managing health conditions [3]. They are also substantially less able to mitigate the financial effects of morbidity through sick pay [7].

### Latin American population of the UK

Stratifying minoritised people into more specific and locally-important categories is important to reveal differential effects of race and ethnicity on health [8–10]. The Latin American community self-identify as having Latin American origin, with a shared culture and language [11, 12]. By aggregating data from the 288 'write-in' ethnic groups of the 2021 Census, 136,062 UK residents identified as having Latin American ethnicity across the White, Other and Mixed higher level groupings [13]. In some areas of London they represent 10% of the population [14]. Despite high levels of tertiary educational attainment, over a third of adults work in elementary jobs in London. They are more likely than other London residents to live in less secure private rented accommodation [15].

There is limited research assessing this community's health needs. 14% of new HIV diagnoses in 2018 were amongst gay and bisexual men (GBM) born in Latin America or the Caribbean [16]. A search of MEDLINE for studies of Latin American health in the UK yielded one qualitative study on the acceptability of HIV self-testing in London [17]. Survey data suggest a high proportion of transnational health seeking and some use of private health services [18]. A mixed methods study of Latin American migrants in London demonstrated difficulties accessing primary care owing to language barriers and perceived discrimination from service providers [19].

We sought to better understand health inequalities affecting the Latin American population compared to other ethnic groups in an urban setting. To do this we developed and validated a phenotype for Latin American ethnicity in routinely-collected data, comparing their life course incidence of locally-important long-term conditions (LTCs) and multiple LTCs (mLTCs) to other ethnic groups.

## Method

### Study design

Retrospective cohort study using Lambeth DataNet, a pseudonymised database derived from the electronic health records of every adult patient registered to a primary care provider in a superdiverse (45% non-White [13]) borough of London between 1st April 2005 and 1st February 2022. There are 40 primary care providers in the sample. This data extraction was performed on 1st February 2022. The authors did not have access to information that could identify individual participants during or after data collection.

For each LTC of interest, individuals were included in a cohort upon registration at a primary care provider or upon turning 18 years of age. They were followed up until an incident LTC, 1st February 2022 (when data collection ceased), deregistration or death. The unit of follow-up was years of age. Individuals were excluded if they were: not resident in London, over 110 years of age; of unknown sex; with a deregistration date after 1st February 2022; with a previous diagnosis of LTC upon cohort registration. Owing to reduced data resolution, participants with less than one year of follow-up were imputed to have six months of follow-up.

### Exposures

Latin American ethnicity was identified in the record using SNOMED-CT codes representing ethnic group, country of birth and main spoken language: (Fig 1). Codelists are available on Figshare [20].

**Self-ascribed ethnic group.** one author (JS) categorised all ethnic groups into confirmed, possible and unlikely to be assigned to Latin American residents. Where more than one ethnic group was recorded we used the most specific (that is, non-Other) group.

**Country of birth.** majority Spanish- and Portuguese-speaking countries in the Americas.

**Main spoken language.** Spanish; Portuguese; indigenous languages spoken by at least 100,000 inhabitants of Latin America according to United Nations estimates [21].

We validated this exposure qualitatively with Indoamerican Refugee and Migrant Organisation, a Latin American advocacy charity. We used linear regression and correlation coefficients to determine the extent to which, for each lower super output area (LSOA), the prevalence of Latin American-born people in the 2011 Census [14] is explained by the proportion of Latin Americans identified in the dataset on 27th March 2011.

We used the aggregated Index of Multiple Deprivation (IMD) 2019 score [22] for each individual, by either 2011 or 2001 LSOA boundaries. The IMD is a weighted score of deprivation for an area with a resident population of between 1,000 and 3,000 covering the domains of education, housing, income, health, crime, housing and the living environment [22]. Each LSOA's IMD score was split into within-borough quintiles.

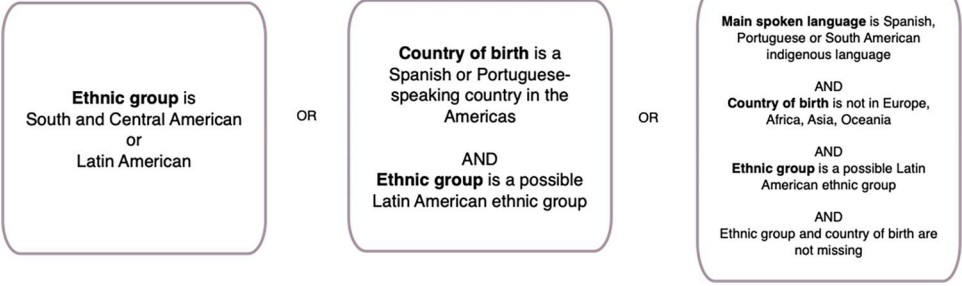

**Fig 1. Logic diagram used to phenotype the Latin American population within primary care records.**

## Outcomes

We used SNOMED-CT codes for 32 LTCs and 4 risk factors: substance use, obesity, alcohol use (ever having recorded as >14 units/week consumption) and hypercholesterolaemia. These were selected by consensus as locally relevant for ethnically diverse urban populations [23]. The clinical codes for each LTC are available on request. The first recorded date of each LTC was considered the date of LTC incidence. To reduce the risk of misclassifying prevalent disease as incidence [24], we excluded individuals with LTC incidence within 6 months of their cohort start date, or 3 months of registration for those registered on 1st April 2005.

## Statistical analysis

We compared sociodemographic characteristics and LTC cumulative incidence for the Latin American and non-Latin American populations. We calculated the cumulative incidence of multimorbidity (having more than one diagnosed LTC on separate dates) for each group. We used cumulative incidence function plots to visualise the differential acquisition of LTCs in the Latin American, Black and White ethnic groups with increasing age.

For each LTC participants were nested within LSOAs and GP practices. We fitted mixed-effects multivariable Fine-Gray regression models to account for competing risks of death [25]. We estimated sub-distribution hazard ratios for the Latin American, compared to the White British, ethnic groups, for each LTC. Each model was adjusted *a priori* for age, sex, smoking status (ever/never), and within-borough IMD quintiles, similarly to other large ethnic disparity studies using UK primary care data [3, 4] We analysed only patients with a recorded ethnic group, and then conducted sensitivity analyses imputing missing ethnic group data as either White British or Latin American. Analysis was performed in R 4.2.1.

## Ethics statement

All data were extracted under the terms of a signed data sharing agreement with each GP practice and with project-specific approval following submission of a data privacy impact assessment, approved by Lambeth Clinical Commissioning Group on 2 November 2017. Information governance approval required 'low number suppression', ensuring that data could not be displayed if the patient number was 10 or less in any given category; in these circumstances, data reporting would state: '≤10 patients'. Separate ethical committee approval was not required (Health Research Authority, 29 September 2017) nor was consent required, since all data were fully anonymised for the purposes of research access and all patient identifiable data had been removed.

## Results

After data cleaning, a total of 890,438 individuals were included in the dataset for analysis (Fig 2). Participants entered the study at a median 30 years of age (interquartile range (IQR) 25–39), with median follow-up of 4.2 years (IQR 1.8–8.7 years). 25,916 (2.9%) individuals died during follow-up.

## Missing data

Ethnicity data was missing for 18% (n = 164673) of the study population; main spoken language data for 31% (n = 274840); country of birth data was missing for 52% (n = 464763). 16% had missing data for all three variables; 14% for two variables; 27% for one variable. 386,413 of 890,438 (43%) participants had a recorded ethnic group, main spoken language, and country of birth.

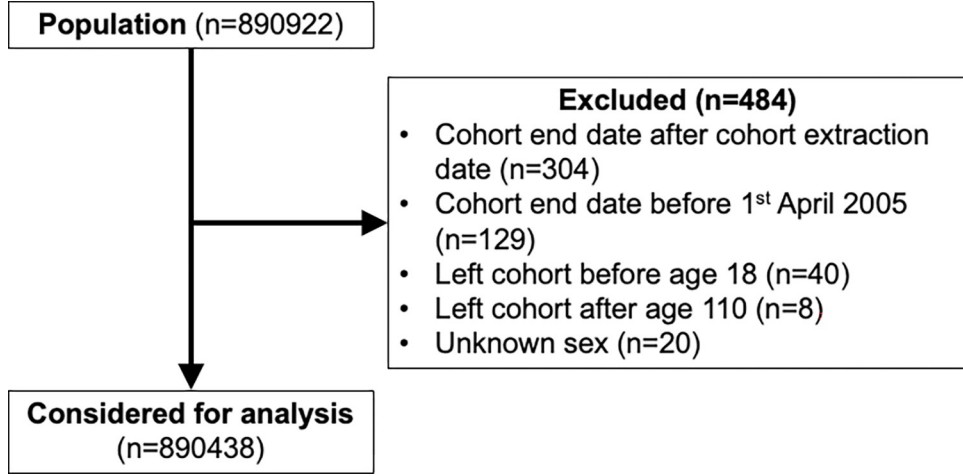

**Fig 2. Flowchart demonstrating the total population considered for analysis.**

## Latin American population

28,617 Latin American individuals were identified: 3.2% of the cohort. The majority (91%) were identified using country codes, with 2,540 more (9%) identified using ethnic group and 49 by language (S1 File). The Latin American population increased from 3,137 in 2005 to 14,048 in 2022 (S1 File). This is 94% of the 2011 Census-estimated population and 29% more than the 10,833 Latin American-identifying Lambeth residents in the 2021 Census [13] (S1 File). Most had SNOMED-CT codes representing the UK Census categories of White (45%), Other (37%) or Mixed (13%) (S1 File).

Latin American people joined the cohort at a median age of 32.6 versus 30.1 for non-Latin Americans (Table 1). Latin Americans were more likely to reside in deprived LSOAs (27% in the most deprived quintile). 22,185 (85%) recorded a non-English main spoken language, with 57% speaking Spanish and 26% Portuguese (Table 1). Compared to the non-Latin American population, Latin Americans were less likely to have ever smoked, exceeded safe limits of alcohol consumption or to report substance use. (Table 1) Total mortality was 0.9% in the Latin American group versus 3% for non-Latin Americans.

## LTCs and mLTCs by ethnic group

In unadjusted analysis, four conditions (systemic lupus erythematosus (SLE), rheumatoid arthritis, HIV/AIDS and hypercholesterolaemia) were more common amongst Latin American people than non-Latin Americans (Fig 3). All other LTCs were less prevalent amongst Latin Americans, with the greatest differences for asthma (4.2% versus 10.6%), anxiety (9.9% versus 15%) and depression (7.1% versus 11.4%). 10.9% of the Latin American group had multimorbidity, compared to 18.0% of non-Latin Americans (S2 File). The predominant LTCs for both groups were chronic pain, anxiety and depression (S2 File).

Cumulative incidence plots describe LTC incidence with increasing age for the Black, White and Latin American ethnic groups (n = 621190) (Fig 4). Latin Americans have similar rates of cancer, morbid obesity and chronic kidney disease to the White ethnic group. For other conditions–atrial fibrillation, osteoarthritis, inflammatory bowel disease–the risk instead matches the Black group and is lower than the White group. There is an earlier age of incident HIV in the Latin American population compared to Black and White populations.

**Table 1. Sociodemographic characteristics of Latin American and non-Latin American patients, 2005–2022.**

| | | Latin American (%) | non-Latin American (%) |
|---|---|---|---|
| Total (n = 890438) | | 28568 (3.2) | 861870 (96.8) |
| Age (n = 890438) | | | |
| | 18–30 | 6348 (22.2) | 242230 (28.1) |
| | 30–45 | 12614 (44.2) | 374163 (43.4) |
| | 45–60 | 7131 (25.0) | 140120 (16.3) |
| | 60–80 | 2200 (7.7) | 79954 (9.3) |
| | 80+ | 275 (1.0) | 25403 (2.9) |
| Sex (n = 890438) | | | |
| | Male | 12916 (45.2) | 414289 (48.1) |
| | Female | 15652 (54.8) | 447581 (51.9) |
| Ethnic group (n = 725765) | | | |
| | White | 11848 (44.8) | 472549 (67.6) |
| | Black/African/Caribbean/Black British | 1332 (5.0) | 120073 (17.2) |
| | Asian/Asian British | 30 (0.1) | 56411 (8.1) |
| | Mixed/Multiple ethnic group | 3438 (13.0) | 31598 (4.5) |
| | Other ethnic group | 9826 (37.1) | 18660 (2.7) |
| Deprivation quintile (1 = most deprived) (n = 890438) | | | |
| | 1 | 7614 (26.7) | 170474 (19.8) |
| | 2 | 7089 (24.8) | 170999 (19.8) |
| | 3 | 5499 (19.2) | 172589 (20.0) |
| | 4 | 4853 (17.0) | 173234 (20.1) |
| | 5 | 3513 (12.3) | 174574 (20.3) |
| Main spoken language (n = 615598) | | | |
| | English | 3930 (15.0) | 454854 (77.2) |
| | Other | 226 (0.9) | 97922 (16.6) |
| | Portuguese | 6945 (26.6) | 20443 (3.5) |
| | Spanish | 15014 (57.5) | 16264 (2.8) |
| Ever smoked (n = 890438) | | | |
| | No | 19387 (67.9) | 506464 (58.8) |
| | Yes | 9181 (32.1) | 355406 (41.2) |
| Excess alcohol use (n = 890438) | | | |
| | None | 28251 (98.9) | 831171 (96.4) |
| | >14 units/week | 103 (0.4) | 10784 (1.3) |
| | Alcohol dependence | 214 (0.7) | 19915 (2.3) |
| Substance use (n = 890438) | | | |
| | None | 28265 (98.9) | 842727 (97.8) |
| | Substance use | 171 (0.6) | 7424 (0.9) |
| | Substance dependence | 132 (0.5) | 11719 (1.4) |

## Multivariable analysis

In multivariable analysis compared to the White British ethnic group (Fig 5, S3 File), HIV/AIDS rates were twice as high among Latin American people (adjusted hazard ratio (HR) 2.00, 95% CI 1.65–2.43). There was twice the rate of SLE (HR 2.28; 95% CI 1.18–4.38), a 70% increased hazard of rheumatoid arthritis (HR 1.69, 1.32–2.17) and 20% higher recorded osteoarthritis (HR 1.22, 1.12–1.33). There were 60% increased rates of diabetes (HR 1.61, 95% CI 1.47–1.77).

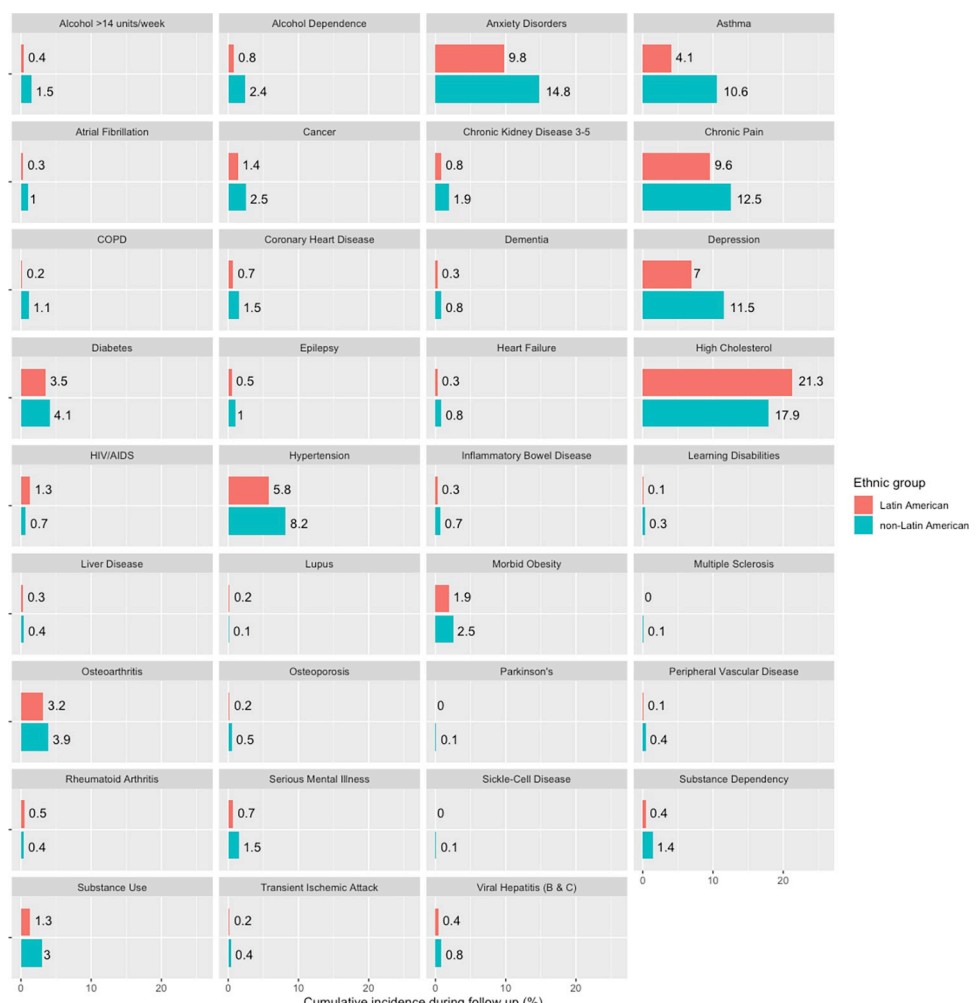

**Fig 3. Cumulative incidence of 37 long-term conditions and risk factors during follow up.** Latin American versus non-Latin American population. n = 890438.

Despite 10% increased rates of hypercholesterolaemia (HR 1.13, 1.09–1.17), there was no evidence of differential rates of stroke or transient ischaemic attack (Fig 5) and there were lower rates of coronary heart disease (HR 0.82, 0.68–1.00). We found statistically strong evidence of lower rates of all cancers (HR 0.75, 0.66–0.85), atrial fibrillation (HR 0.50, 0.39–0.63), and chronic kidney disease stages 3–5 (HR 0.75, 0.65–0.87).

Rates of recorded anxiety and depression were a quarter lower in Latin American people (95% CIs 0.73–0.81 and 0.71–0.81 respectively), with no differential incidence of serious mental illness. Sensitivity analyses (S3 File) demonstrated that the direction of each effect estimate was robust to imputations of ethnic group.

## Discussion

### Summary

This study describes the incidence of common LTCs in the Latin American population.–a minoritised group who identify across several census ethnic groups. We adjusted for common confounders, providing data to address their disproportionate health challenges. We provide a

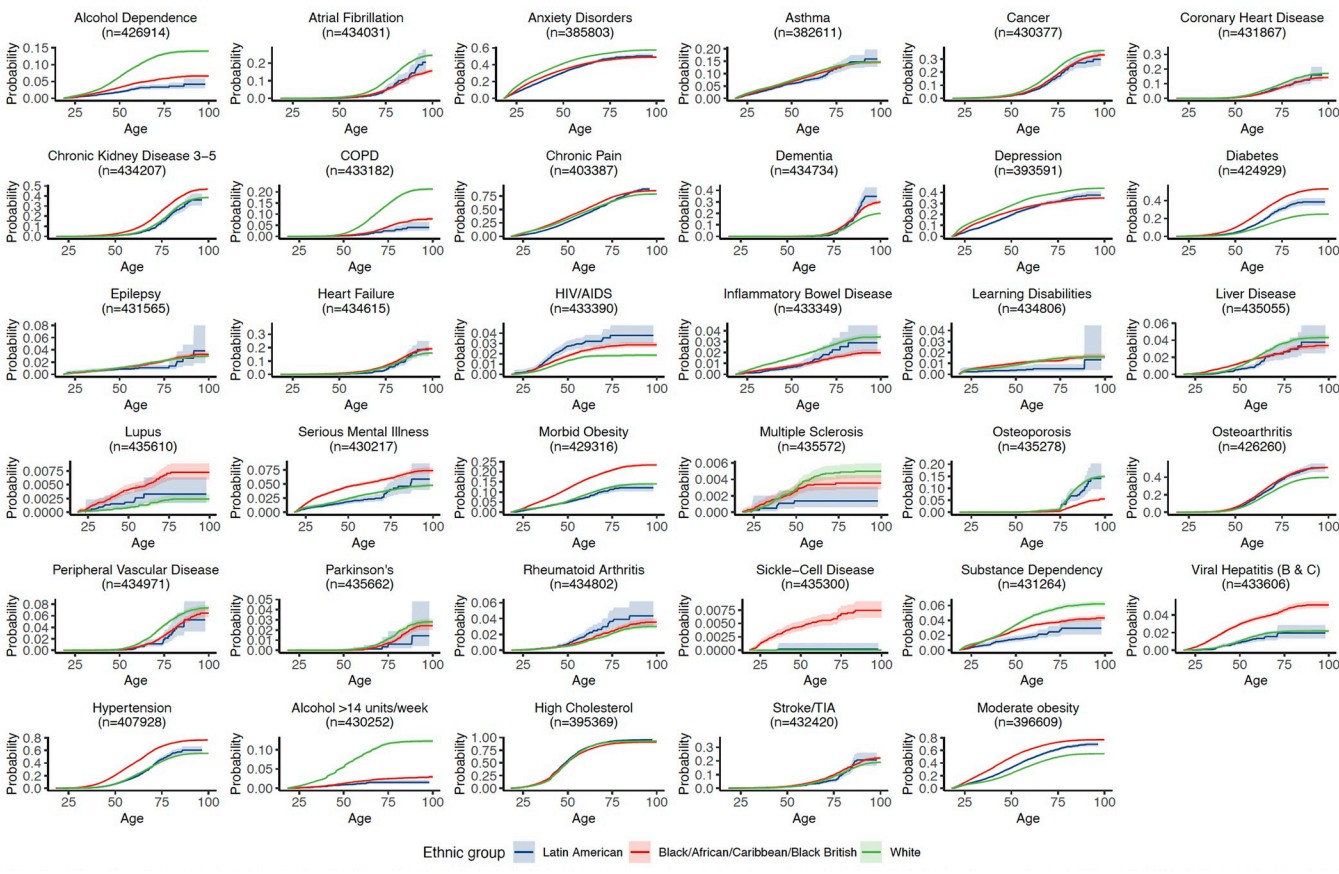

**Fig 4. Cumulative incidence function plots for each long-term condition for Latin American-, Black- and White-identifying patients.** Shaded areas represent 95% confidence intervals for each curve.

reproducible methodology to describe health inequalities in LTC diagnoses for minoritised populations.

Latin Americans are at markedly higher risk of developing HIV/AIDS, with higher incidence than the Black ethnic group. This is in the context of Lambeth having the highest HIV prevalence in Europe (1.3%) [26], and much lower prevalences in most Latin American countries [27]. As 18% of gay and bisexual men newly diagnosed with HIV in 2018 were Latin American [28], this strengthens evidence for HIV transmission occurring in a key population in London [29].

The Latin American community compared to the White ethnic group has increased obesity, diabetes and hypercholesterolaemia rates; other risk factors (smoking, high alcohol intake recording) are less prevalent. low smoking and recorded alcohol consumption. Despite similar rates of hypertension in the White British and Latin American groups, there were lower risks of its cardiovascular sequelae. The markedly lower rates of chronic obstructive pulmonary disease could represent differing occupational exposures or differential measurement of smoking status by general practitioners [30].

Despite national surveys indicating high levels of self-reported anxiety scores in the Mixed and Other populations [31], there were lower rates of anxiety and depression in this cohort which may represent underreporting. Musculoskeletal conditions are more likely to be diagnosed in the Latin American than White British population.

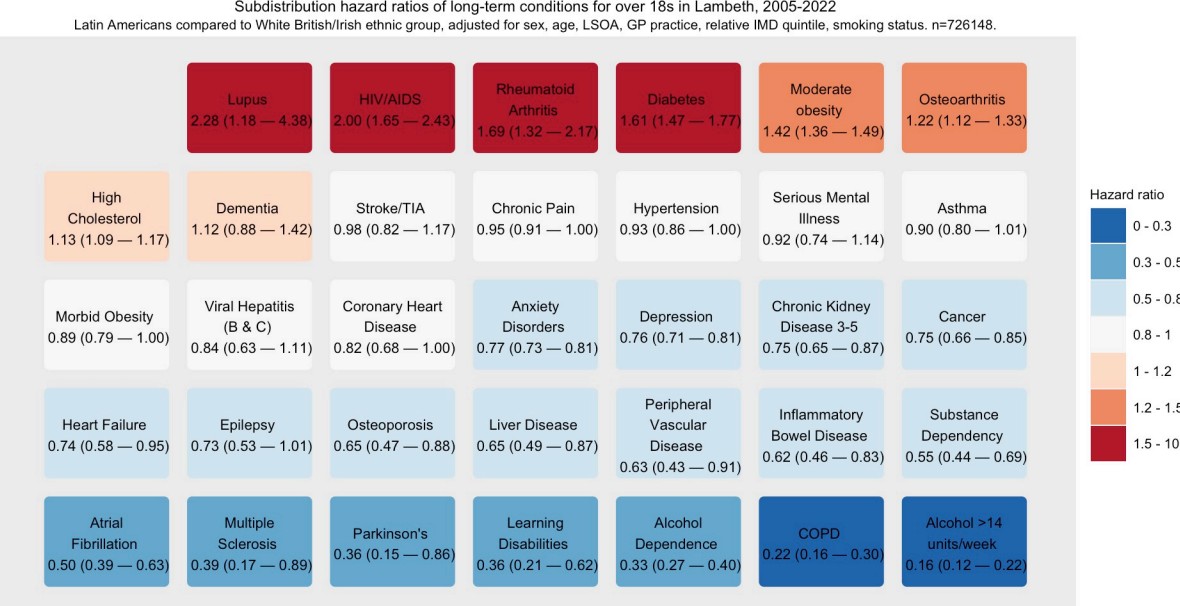

**Fig 5. Sub-distribution hazard ratios and 95% confidence intervals for long-term conditions and risk factors for Latin American patients compared to the White ethnic group.** Owing to small numbers of sickle cell disease in Latin American patients an estimate was not possible. Competing risks regression adjusted for age, sex, lower super output area, deprivation quintile, smoking status. n = 726148.

## Strengths and limitations

In London one-fifth of Latin American survey respondents have not used primary care [32], yet we were able to identify numbers comparable to the 2011 Census and more than the 2021 Census. We identified this many Latin American individuals notwithstanding missing ethnic group information for one fifth of individuals and higher levels of missingness for country of birth and language data. The Latin American phenotype was robust to validation using both country of birth data (2011 Census) and the 288 'write-in' ethnic group categories of the 2021 Census. The dataset is inclusive and representative of the urban population studied. As this study used data from every care provider in a defined area, case ascertainment of many LTCs is likely to be high. HIV status in particular is well-recorded in this dataset owing to long-established opt-out testing in primary care and emergency departments [33]. Other conditions–risk factors, osteoarthritis, depression and anxiety–may be differentially measured between ethnic groups and therefore be prone to bias in either direction. Case ascertainment for some LTCs may be improved by a linkage to secondary care data. The results were robust to sensitivity analyses. Although results may not be generalisable to rural and less ethnically dense areas, many structural factors affecting Latin American health may apply in other UK urban areas.

We adjusted for common biomedical and sociodemographic variables in line with other studies on ethnic differences in health conditions [5, 6]. However residual confounding is inevitable, as Latin American people are subject to the unmeasured and intersectional effect of structural, neighbourhood-level, practice-level, and patient-level determinants of health [34, 35]. Language preferences may change over time as migrants become acculturated in the UK; this cannot be accounted for. Finally definitions of race and ethnicity change over time [8] and were not longitudinally recorded in these data. Including other dimensions of ethnicity such as religion and migrant status [9] might produce a more specific exposure definition. This study was not able to capture individual-level socioeconomic data, which may underestimate neighbourhood and individual determinants of health [36].

Geographical areas with more Latin American residents tended to have higher-than-expected representation in the dataset, when compared to census data. This was sometimes higher than the average 6% General Practice (GP) over-registration in London [37]. Further research should identify whether this represents underestimation in the 2021 census, or emigration without deregistration from GP practices. If emigration without deregistration is common this would be an immortal time bias which could underestimate the effect of migrant group status on LTC incidence [38].

## Comparison with existing literature

Hispanic Americans have four times the rate of HIV infection, compared to non-Hispanic White Americans [39]. We found 40% increased rates of diabetes, compared to 65% increased rates of diabetes reported in US Hispanic populations [40]. The higher rates of lupus and rheumatoid arthritis are also consistent with US data, however given the pathophysiological diversity of SLE [41] there may be both genetic and structural factors at play [41–44]. Structural factors affecting the UK Latin American community are likely to be specific to the UK, limiting the ability to compare our findings to those of Hispanic migrants elsewhere.

Low levels of harmful alcohol consumption amongst Latin Americans are also found in US literature, where information on alcohol is not systematically collected [45]. Although there is no ethnic disparity in recording of alcohol consumption [46], it is not clear whether language preference may affect the recording of risk factor information in primary care.

## Conclusion

This work provides a reproducible methodology to describe chronic health disparities in minoritised urban populations over time, after adjustment for common confounding factors. Here we describe especially high rates of HIV/AIDS amongst the Latin American community in London, as well as increased rates of rheumatological disorders and diabetes. Further research should include longitudinal multimorbidity clusters [47] and care quality, from both primary care and secondary care, to better identify health system factors that affect ethnic inequalities in health.

## Supporting information

**S1 File. Additional tables and figures supporting the validation of the Latin American cohort.**
(DOCX)

**S2 File. Multiple long-term conditions in the Latin American cohort.**
(DOCX)

**S3 File. Multivariable regression coefficients and sensitivity analyses for long-term conditions in the Latin American population.**
(DOCX)

## Acknowledgments

The authors would like to acknowledge the contribution of the Indoamerican Refugee and Migrant Organization for their valuable help in validating the codelists used in this study.

## Author Contributions

**Conceptualization:** James Scuffell, James Bailey, Hiten Dodhia, Mark Ashworth.

**Data curation:** James Scuffell, Stevo Durbaba.

**Formal analysis:** James Scuffell, Mark Ashworth.

**Funding acquisition:** James Scuffell, Mark Ashworth.

**Investigation:** James Scuffell, Mark Ashworth.

**Methodology:** James Scuffell, James Bailey, Hiten Dodhia, Mark Ashworth.

**Project administration:** James Scuffell, Stevo Durbaba, Mark Ashworth.

**Resources:** James Scuffell.

**Software:** Stevo Durbaba.

**Supervision:** Hiten Dodhia, Mark Ashworth.

**Validation:** James Scuffell.

**Visualization:** James Scuffell.

**Writing – original draft:** James Scuffell.

**Writing – review & editing:** James Scuffell, James Bailey, Hiten Dodhia, Stevo Durbaba, Mark Ashworth.

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
