## [Decision Letter · Decision Letter 0]

1 Apr 2024

PONE-D-24-04965Incidence of long-term conditions in the Latin American community of London: a validation and retrospective cohort study of 890,922 primary care records, 2005-2022.PLOS ONE

Dear Dr. Scuffell,

Thank you for submitting your manuscript to PLOS ONE. After careful consideration, we feel that it has merit but does not fully meet PLOS ONE’s publication criteria as it currently stands. Therefore, we invite you to submit a revised version of the manuscript that addresses the points raised during the review process.

We look forward to receiving your revised manuscript.

Kind regards,

Doaa Mohamed Attia, MBBCH

Academic Editor

PLOS ONE

“JS is funded by a National Institute for Health Research In-Practice Fellowship (NIHR303520). JB is funded by a National Institute for Health Research Doctoral Research Fellowship (NIHR302551). The views expressed are those of the author(s) and not necessarily those of the NIHR or the Department of Health and Social Care.”

3. In the online submission form you indicate that your data is not available for proprietary reasons and have provided a contact point for accessing this data. Please note that your current contact point is a co-author on this manuscript. According to our Data Policy, the contact point must not be an author on the manuscript and must be an institutional contact, ideally not an individual. Please revise your data statement to a non-author institutional point of contact, such as a data access or ethics committee, and send this to us via return email. Please also include contact information for the third party organization, and please include the full citation of where the data can be found.

3. We note that Figure S4 in your submission contain [map/satellite] images which may be copyrighted. All PLOS content is published under the Creative Commons Attribution License (CC BY 4.0), which means that the manuscript, images, and Supporting Information files will be freely available online, and any third party is permitted to access, download, copy, distribute, and use these materials in any way, even commercially, with proper attribution. For these reasons, we cannot publish previously copyrighted maps or satellite images created using proprietary data, such as Google software (Google Maps, Street View, and Earth). For more information, see our copyright guidelines: http://journals.plos.org/plosone/s/licenses-and-copyright.

1. You may seek permission from the original copyright holder of Figure S4 to publish the content specifically under the CC BY 4.0 license. 

Reviewers' comments:

Reviewer's Responses to Questions

**Comments to the Author**

1. Is the manuscript technically sound, and do the data support the conclusions?

Reviewer #1: Yes

Reviewer #2: No

2. Has the statistical analysis been performed appropriately and rigorously? 

Reviewer #1: Yes

Reviewer #2: Yes

3. Have the authors made all data underlying the findings in their manuscript fully available?

Reviewer #1: Yes

Reviewer #2: No

4. Is the manuscript presented in an intelligible fashion and written in standard English?

Reviewer #1: Yes

Reviewer #2: Yes

5. Review Comments to the Author

Reviewer #1: Background Review:

The manuscript addresses an important gap in healthcare research by focusing on the health needs of minoritized populations, specifically the Latin American community in the United Kingdom. The background provides relevant context regarding the underrepresentation of Latin American individuals in health studies and highlights the diversity within this population. However, further discussion on the social determinants of health affecting Latin American communities in the UK could enrich the background section.

Aim Review:

The aim of the study is clearly articulated, focusing on characterizing the incidence of long-term conditions (LTCs) and multiple LTCs (mLTCs) among the Latin American community in London. The research question is well-defined and addresses a significant public health issue, particularly in the context of superdiverse urban areas. However, providing more detail on the specific LTCs of interest and their relevance to the Latin American population would enhance the clarity of the aim.

Design and Setting Review:

The manuscript describes a retrospective cohort study utilizing pseudonymized primary care data from a large urban area of London spanning from 2005 to 2022. The choice of study design and setting is appropriate for capturing longitudinal health data and understanding healthcare utilization patterns among diverse populations. However, providing information on the primary care practices included in the study and any potential limitations associated with the use of electronic health records would strengthen the design and setting section.

Method Review:

The methods section outlines the identification of Latin American individuals using country of birth, language, and ethnicity codes, validated against Census findings. The use of multivariable competing risks regression models to estimate the effect of ethnicity on LTC incidence is appropriate and statistically rigorous. However, providing details on how missing data or misclassification of LTCs were addressed in the analysis would enhance the transparency of the methodology.

Results Review:

The results section presents key findings regarding LTC incidence among the Latin American population in London. The use of hazard ratios and confidence intervals effectively summarizes the associations between ethnicity and LTCs. However, providing absolute incidence rates or prevalence estimates for each LTC would enhance the interpretation of results and facilitate comparisons with other studies.

Conclusion Review:

The conclusion effectively summarizes the study's findings and emphasizes the importance of targeted health interventions for the Latin American community in London. The implications for public health practice and policy are clearly articulated, highlighting the potential for using primary care data to inform equitable health interventions. However, acknowledging potential limitations of the study, such as the reliance on primary care data and potential selection bias, would strengthen the conclusion's validity.

Overall, the manuscript contributes valuable insights into the health disparities experienced by the Latin American community in London and the need for tailored healthcare interventions. Strengthening the background with additional context, providing more detail on LTCs of interest, addressing potential limitations in the methodology, and acknowledging study limitations in the conclusion would enhance the manuscript's overall quality and impact.

Reviewer #2: Reviewer comments PONE-D- 24-04965

Thank you for the opportunity to review the authors’ work.

The authors introduce the subject of their study by providing an argument for the relevance of this work for other follow up studies.

Aim: The aim presented in the abstract differs in meaning to the aim stated at the end of the Introduction. One talks about describing the profile of long-term conditions in the Latin American community while the other also plans to describe health inequalities evidenced by the prevailing conditions in the Latin American community.

Methodology: A retrospective review of pseudonymized health data collated from GP practices is inadequate to effectively describe health inequalities as it does not collect enough information in this study to answer these questions.

Results and Discussion: The authors provide a number of tables with adequate statistical analysis using multivariate regression analysis. The findings describe prevailing health conditions in the Latin American population in this community. The efforts of the authors to stretch the findings to describing health inequalities is inadequate as they do not have enough information and several confounding factors make it impossible to do so.

Strength and Limitations: The authors have described in some detail the many limitations to this study. The very reasons the conclusion is not adequate is described in this section, as they highlight the challenges of missing data, confounders and not enough information collated based on the study method.

The conclusion: I recommend the authors conclude on the point of describing the profile of illnesses in the different ethnic groups identified in the study, the study is not powered enough to conclude on the health inequalities experienced by the Latin American population in comparison to other ethnic groups in the study population.

Reference: This is done correctly in the Vancouver style as required by the journal. The literature presented is recent and relevant to the population where the study was done.

Please see manuscript for comments on the sections.

Thank you.

Reviewer.

6. PLOS authors have the option to publish the peer review history of their article (what does this mean?). If published, this will include your full peer review and any attached files.

Reviewer #1: **Yes: **Doaa Attia

Reviewer #2: No

---

## [Author Response · Author response to Decision Letter 0]

8 Apr 2024

Dear Dr Attia,

Thank you once again for your consideration of our manuscript ‘Incidence of long-term conditions in the Latin American community of London: a validation and retrospective cohort study of 890,922 primary care records, 2005-2022’. We are very grateful for the additional comments from peer reviewers who have very much helped to strengthen the manuscript. We have responded to each of the comments from peer reviewers in turn below. 

I have also updated the filenames and author page according to the PLOS One guidance.

Thank you for noticing the licensing agreement for figure S4 in the text. I do apologise; the OpenStreetMap license we had quoted referred to its documentation rather than the map itself. The map is actually licensed for use under the Open Data Commons’ Open Database License, whose terms are compatible with the CCAL CC BY 4.0 licence. I have adapted the S1 File accordingly and uploaded a revised version of this file.

Yours sincerely on behalf of the co-authors,

Dr Jamie Scuffell

Reviewer #1

Background Review:

The manuscript addresses an important gap in healthcare research by focusing on the health needs of minoritized populations, specifically the Latin American community in the United Kingdom. The background provides relevant context regarding the underrepresentation of Latin American individuals in health studies and highlights the diversity within this population. However, further discussion on the social determinants of health affecting Latin American communities in the UK could enrich the background section.

Many thanks for these comments. We have further contextualised the social determinants of health for the Latin American community by adding additional information on employment and educational status to lines 69-71 of the manuscript. 

Lines 69-71 now read:

Despite high levels of tertiary educational attainment, over a third of adults work in elementary jobs in London. They are more likely than other London residents to live in less secure private rented accommodation. [15]

Lines 77-79 read:

A mixed methods study of Latin American migrants in London demonstrated difficulties accessing primary care owing to language barriers and perceived discrimination from service providers. [19]

Aim Review:

The aim of the study is clearly articulated, focusing on characterizing the incidence of long-term conditions (LTCs) and multiple LTCs (mLTCs) among the Latin American community in London. The research question is well-defined and addresses a significant public health issue, particularly in the context of superdiverse urban areas. However, providing more detail on the specific LTCs of interest and their relevance to the Latin American population would enhance the clarity of the aim.

Thank you. We have explained in the ‘outcomes’ section of the methods that the LTCs chosen were based on consensus exercises with local health stakeholders. We have clarified this further by describing the LTCs as ‘locally-important LTCs’ in line 82.

Line 81: ‘To do this we developed and validated a phenotype for Latin American ethnicity in routinely-collected data, comparing their life course incidence of locally-important long-term conditions (LTCs) to other ethnic groups.’

Design and Setting Review:

The manuscript describes a retrospective cohort study utilizing pseudonymized primary care data from a large urban area of London spanning from 2005 to 2022. The choice of study design and setting is appropriate for capturing longitudinal health data and understanding healthcare utilization patterns among diverse populations. However, providing information on the primary care practices included in the study and any potential limitations associated with the use of electronic health records would strengthen the design and setting section.

Thank you for your thoughts on how to better describe the dataset. We had not made it clear that this is a sample of every adult registered to a primary care provider in a borough of London. 

We have clarified the study sample in lines 86-89:

Retrospective cohort study using Lambeth DataNet, a pseudonymised database derived from the electronic health records of every adult patient registered to a primary care provider in a superdiverse (45% non-White[13]) borough of London between 1st April 2005 and 1st February 2022. There are 40 primary care providers in the sample.

Method Review:

The methods section outlines the identification of Latin American individuals using country of birth, language, and ethnicity codes, validated against national Census findings. The use of multivariable competing risks regression models to estimate the effect of ethnicity on LTC incidence is appropriate and statistically rigorous. However, providing details on how missing data or misclassification of LTCs were addressed in the analysis would enhance the transparency of the methodology.

Thank you. We explain in the methods that we conducted a sensitivity analysis for the exposure which was most likely to cause misclassification bias: missing ethnic group data. We have addressed the risk of misclassification of LTCs in an additional section in the discussion section of the paper. 

Lines 259-264 now read:

As this study used data from every care provider in a defined area, case ascertainment of many LTCs is likely to be high. HIV status in particular is well-recorded in this dataset owing to long-established opt-out testing in primary care and emergency departments. [33] Other conditions – risk factors, osteoarthritis, depression and anxiety – may be differentially measured between ethnic groups and therefore be prone to bias in either direction. Case ascertainment for some LTCs may be improved by a linkage to secondary care data.

Results Review:

The results section presents key findings regarding LTC incidence among the Latin American population in London. The use of hazard ratios and confidence intervals effectively summarizes the associations between ethnicity and LTCs. However, providing absolute incidence rates or prevalence estimates for each LTC would enhance the interpretation of results and facilitate comparisons with other studies.

Thank you. We do describe cumulative incidence rates for each LTC in the Results section (lines 185–191). Given that the absolute incidence rates for LTCs will differ substantially by age, we have instead presented cumulative incidence function curves in Figures 3 and 4. 

Conclusion Review:

The conclusion effectively summarizes the study's findings and emphasizes the importance of targeted health interventions for the Latin American community in London. The implications for public health practice and policy are clearly articulated, highlighting the potential for using primary care data to inform equitable health interventions. However, acknowledging potential limitations of the study, such as the reliance on primary care data and potential selection bias, would strengthen the conclusion's validity.

Thanks for these comments. As this is observational data we recognise the potential selection bias that can arise from using primary care data. However, as demonstrated in the study, a large proportion of Latin American residents of the borough of Lambeth – approximately 90% – are registered at a primary care practice. As this is the only means of accessing publicly-funded health services in the United Kingdom, there is likely good case ascertainment of these LTCs. We have further discussed this with additions to the limitations section., which now read:

Lines 259-264 now read:

As this study used data from every care provider in a defined area, case ascertainment of many LTCs is likely to be high. HIV status in particular is well-recorded in this dataset owing to long-established opt-out testing in primary care and emergency departments. [33] Other conditions – risk factors, osteoarthritis, depression and anxiety – may be differentially measured between ethnic groups and therefore be prone to bias in either direction. Case ascertainment for some LTCs may be improved by a linkage to secondary care data.

Overall, the manuscript contributes valuable insights into the health disparities experienced by the Latin American community in London and the need for tailored healthcare interventions. Strengthening the background with additional context, providing more detail on LTCs of interest, addressing potential limitations in the methodology, and acknowledging study limitations in the conclusion would enhance the manuscript's overall quality and impact.

Thank you for this feedback.

Reviewer #2

Reviewer 2 also offered specific feedback in comments on the manuscript and this has been very helpful to improve these areas of the manuscript. We have incorporated responses into this text and directly into the text of the manuscript.

Thank you for the opportunity to review the authors’ work.

The authors introduce the subject of their study by providing an argument for the relevance of this work for other follow up studies.

Aim: The aim presented in the abstract differs in meaning to the aim stated at the end of the Introduction. One talks about describing the profile of long-term conditions in the Latin American community while the other also plans to describe health inequalities evidenced by the prevailing conditions in the Latin American community.

Thank you. Our aim was to describe some health inequalities faced by the Latin American community by considering differential incidence of long-term conditions by ethnicity. The LTCs we chose were chosen as being locally relevant. We have clarified this, as you have suggested, by focusing on incident conditions arising among the Latin American population (lines 79-81). Responding to the comment attached to lines 76-77, we have also clarified the aim in the Abstract so that it matches that in the introduction.

Lines 79-81 now read:

We sought to better understand health inequalities affecting the Latin American population compared to other ethnic groups in an urban setting. To do this we developed and validated a phenotype for Latin American ethnicity in routinely-collected data, comparing their life course incidence of locally-important long-term conditions (LTCs) and multiple LTCs (mLTCs) to other ethnic groups.

The aim in the Abstract now matches this:

Characterise the incidence of long-term conditions (LTCs) and multiple LTCs (mLTCs) amongst the Latin American community of London. Compare the incidence of LTCs in the Latin American population to other ethnic groups.

Methodology: A retrospective review of pseudonymized health data collated from GP practices is inadequate to effectively describe health inequalities as it does not collect enough information in this study to answer these questions.

Many thanks for your comments. We agree with your concerns of demonstrating causality from observational data such as primary care, which is prone to selection bias and residual confounding. However in multivariable analysis we were able to adjust for what are likely the most substantial confounders (age, sex, neighbourhood-level deprivation and between-practice variation in diagnosis). This is in keeping with the methodologies of other large-scale ethnic inequality studies, such as Mathur et al. (2021). We have added an explicit reference to this in the methods section (line 143):

 Each model was adjusted a priori for age, sex, smoking status (ever/never), and within-borough IMD quintiles, similarly to other large ethnic disparity studies using UK primary care data[3,4]. 

We discuss the limitations of this in the discussion section, which we have further expanded in lines 259-264 as a result of your review.

Lines 259-264 now read:

As this study used data from every care provider in a defined area, case ascertainment of many LTCs is likely to be high. HIV status in particular is well-recorded in this dataset owing to long-established opt-out testing in primary care and emergency departments. [33] Other conditions – risk factors, osteoarthritis, depression and anxiety – may be differentially measured between ethnic groups and therefore be prone to bias in either direction. Case ascertainment for some LTCs may be improved by a linkage to secondary care data.

Studies based on UK primary care data have been shown to be representative of the overall UK population (Wolf et al. 2019). This study includes every patient registered to every practice in a borough of London and includes small-area neighbourhood deprivation indicators. We did not aim in this study to demonstrate causality. We have elaborated further on how UK deprivation indicators are calculated to make it clearer to international audiences how socioeconomic conditions are adjusted for in the analysis.

Lines 120-122 now read:

The IMD is a weighted score of deprivation for an area with a resident population of between 1,000 and 3,000 covering the domains of education, housing, income, health, crime, housing and the living environment.

Mathur R, Rentsch CT, Morton CE, Hulme WJ, Schultze A, MacKenna B, Eggo RM, Bhaskaran K, Wong AYS, Williamson EJ, Forbes H, Wing K, McDonald HI, Bates C, Bacon S, Walker AJ, Evans D, Inglesby P, Mehrkar A, Curtis HJ, DeVito NJ, Croker R, Drysdale H, Cockburn J, Parry J, Hester F, Harper S, Douglas IJ, Tomlinson L, Evans SJW, Grieve R, Harrison D, Rowan K, Khunti K, Chaturvedi N, Smeeth L, Goldacre B; OpenSAFELY Collaborative. Ethnic differences in SARS-CoV-2 infection and COVID-19-related hospitalisation, intensive care unit admission, and death in 17 million adults in England: an observational cohort study using the OpenSAFELY platform. Lancet. 2021 May 8;397(10286):1711-1724. doi: 10.1016/S0140-6736(21)00634-6. Epub 2021 Apr 30. 

Achim Wolf, Daniel Dedman, Jennifer Campbell, Helen Booth, Darren Lunn, Jennifer Chapman, Puja Myles, Data resource profile: Clinical Practice Research Datalink (CPRD) Aurum, International Journal of Epidemiology, Volume 48, Issue 6, December 2019, Pages 1740–1740g, https://doi.org/10.1093/ije/dyz034

Results and Discussion: The authors provide a number of tables with adequate statistical analysis using multivariate regression analysis. The findings describe prevailing health conditions in the Latin American population in this community. The efforts of the authors to stretch the findings to describing health inequalities is inadequate as they do not have enough information and several confounding factors make it impossible to do so.

Many thanks. In this paper we did not want to estimate the causal effect of Latin American ethnicity on incidence of LTCs. Rather we wanted to provide pragmatic data and a reproducible methodology for policy-makers to identify ethnic disparities using primary care data. We have clarified this further in the Discussion (lines 235-236) which now read: ‘We provide a reproducible methodology to describe health inequalities in LTC diagnoses for minoritised populations.’

In the study we provide both data on prevailing conditions for the Latin American population, as well as multivariable relative incidence rates with respect to ethnicity. As a result of your comments we have changed the wording to make it more explicit that this is a descriptive study and not a causal one (line 233, 307-310). 

Line 233: ‘This study describes the incidence of common LTCs in the Latin American population. – a minoritised group who identify across several census ethnic groups.’

Lines 307-310: 

This work provides a reproducible methodology to describe chronic health disparities in minoritised urban populations over time, after adjustment for common confounding factors. Here we describe especially high rates of HIV/AIDS amongst the Latin American community in London, as well as increased rates of rheumatological disorders and diabetes.

Strength and Limitations: The authors have described in some detail the many limitations to this study. The very reasons the conclusion is not adequate is described in this section, as they highlight the challenges of missing data, confounders and not enough information collated based on the study method.

In this revision we have further expanded the strengths and limitations section to encompass your comments

Lines 259-264 now rea

---

## [Decision Letter · Decision Letter 1]

4 Oct 2024

Incidence of long-term conditions in the Latin American community of London: a validation and retrospective cohort study of 890,922 primary care records, 2005-2022.

PONE-D-24-04965R1

Dear Dr. Scuffell,

We’re pleased to inform you that your manuscript has been judged scientifically suitable for publication and will be formally accepted for publication once it meets all outstanding technical requirements.

Kind regards,

Angelo Moretti, Ph.D.

Academic Editor

PLOS ONE

Additional Editor Comments (optional):

Reviewers' comments:

Reviewer's Responses to Questions

**Comments to the Author**

1. If the authors have adequately addressed your comments raised in a previous round of review and you feel that this manuscript is now acceptable for publication, you may indicate that here to bypass the “Comments to the Author” section, enter your conflict of interest statement in the “Confidential to Editor” section, and submit your "Accept" recommendation.

Reviewer #3: All comments have been addressed

2. Is the manuscript technically sound, and do the data support the conclusions?

Reviewer #3: Yes

3. Has the statistical analysis been performed appropriately and rigorously? 

Reviewer #3: Yes

4. Have the authors made all data underlying the findings in their manuscript fully available?

Reviewer #3: Yes

5. Is the manuscript presented in an intelligible fashion and written in standard English?

Reviewer #3: Yes

6. Review Comments to the Author

Reviewer #3: The study highlights the incidence of long-term conditions in the Latin American community of London. It is debatable whether the number of participants should be mentioned in the title of the manuscript, however it should be preferred to not include the number of participants in the title.

7. PLOS authors have the option to publish the peer review history of their article (what does this mean?). If published, this will include your full peer review and any attached files.

Reviewer #3: No

---

## [Editor Report · Acceptance letter]

17 Oct 2024

PONE-D-24-04965R1 

PLOS ONE

Dear Dr. Scuffell, 

I'm pleased to inform you that your manuscript has been deemed suitable for publication in PLOS ONE. Congratulations! Your manuscript is now being handed over to our production team.

Kind regards, 

on behalf of

Dr. Angelo Moretti 

Academic Editor

PLOS ONE